# Stock Movement Prediction Using Machine Learning Based on Technical Indicators and Google Trend Searches in Thailand

**Kittipob Saetia [1]** and **Jiraphat Yokrattanasak [2,3,*]**

1   KMITL Digital Analytics and Intelligence Center, School of Science, King Mongkut's Institute of Technology Ladkrabang, Bangkok 10520, Thailand
2   Department of Mathematics, School of Science, King Mongkut's Institute of Technology Ladkrabang, Bangkok 10520, Thailand
3   Business Innovation and Investment Laboratory (B2I-Lab), School of Science, King Mongkut's Institute of Technology Ladkrabang, Bangkok 10520, Thailand
*   Correspondence: jiraphat.yo@kmitl.ac.th

**Abstract:** Machine learning for stock market prediction has recently been popular for identifying stock selection strategies and providing market insights. In this study, we adopted machine learning algorithms to analyze technical indicators, and Google Trends search terms based on the Thai stock market. This study uses three datasets, which are technical indicators, Google Trends search terms, and a combination of the two. The objectives were to study and identify the factors in stock selection, develop and evaluate portfolio selection models using keyword proxies from the three datasets mentioned, and compare the performance of the selected algorithms. In the prediction process, we discovered that the combination of technical indicators and Google Trends search terms while applying Logistic Regression, Random Forest, and Extreme Gradient Boosting (XGBoost) exhibited the highest ROC curves. For success prediction rate and annualized return, Random Forest and XGBoost were almost similar but still different. While XGBoost performs well during a period of market critical conditions (COVID-19), Random Forest performs marginally better than XGBoost during normal market conditions in terms of average success rate.

**Keywords:** stocks; Google Trends; machine learning

## 1. Introduction

An inefficient market is observed when the price of a security at a given point in time does not represent the whole worth of the asset in the stock market. This might be due to investor decision-making behaviors that cannot be predicted using existing data and indicators. It would be beneficial if analysts and investors could grasp the link between variables that cause different market phenomena such as trade volume, online search trends, and investor behaviors. The study and comprehension of these variables will be essential for modeling to more accurately predict the trading behaviors of investors and market movements.

In the context of the investment market, irrational investor behavior can take various forms. One of the examples is investors who are exiting the market out of fear of a potential market downturn. Future asset price fluctuations and the outlook for the market as a whole are influenced by behavior patterns. To date, prior studies have investigated issues related to modeling and stock prediction by understanding human behaviors and their impacts on the stock markets. Economic and technical indicators, market information, headline news, and online search terms were incorporated into these research studies (Antonio Agudelo Aguirre et al. 2020; Atkins et al. 2018; Chen et al. 2020; Dash and Dash 2016; Huang et al. 2019; Li et al. 2014; Papadamou et al. 2022; Poutachidou and Papadamou 2021; Teixeira and de Oliveira 2010; Yu et al. 2013). Most of the mentioned indicators were proven to be helpful in predicting the total price of shares or securities. However, relatively few

efforts are made to forecast the price per security or price per share, especially in low- and middle-income country settings.

In a related study, Teixeira and de Oliveira (2010) combined technical indicators, such as Simple Moving Average (SMA), Relative Strength Index (RSI), Stochastic Oscillator K, Stochastic Oscillator D, and Bollinger bands together with the K-Nearest Neighbor Algorithm (KNN) to create a prediction model. This study concluded that the method outperformed the buy-and-hold strategy for 12 out of the 15 Brazil equities taken into account in the studies when we compare the outcomes to those. As a result, this study agreed that it is possible to use this method to forecast actual short-term stock trends.

Another study by Dash and Dash (2016) in India using technical indicators, but combined with machine learning, was tested in the S&P 500 and BSE SENSEX contexts. Among the many modeling methodologies, the Artificial Neural Network (ANN) generated the largest returns, with rises of 34.42% and 42.58% in the BSE SENSEX and S&P 500, respectively.

Scholars also tried to leverage unstructured qualitative data to predict stock prices and returns in capital markets. Li et al. (2014) adapted the headline news, blogs, and financial discussion board coupled with the Support Vector Regression (SVR). The study concluded that the impact of the media on companies varied by the nature of the information. The fundamentals of an article, such as company-specific news, can enhance investors' knowledge. Public sentiment can cause volatility in the mood of investors and interferes with investor decision-making. On the other hand, Atkins et al. (2018) combined two types of information, news from Reuters USA and financial data on Yahoo Finance incorporated into the Latent Dirichlet Allocation. This study concluded that news data influence volatility forecasts better than stock market closing prices forecasts.

The review of stock market forecasting processes conducted by Bustos and Pomares-Quimbaya (2020) shed light on the overall process of forecasting stock market movements using machine learning (ML), deep learning, text mining, and clustering techniques to create an investment model. These techniques, used as investors' strategy planning and decision-making tools, tend to perform better than traditional trading strategies. Evidently, Huang et al. (2019) investigated the effectiveness of stock prediction models for the S&P 500 Index with Google Trends, Support Vector Machine (SVM), Ridge Regression, Lasso Regression, and Elastic-Net Regression. With 63.75% accuracy, the top-performing model was Ridge Regression with selection factor modifying VAR model. Papadamou et al. (2022) examined the connections between investor sentiment, as represented by Google Trends, and stock market return, volatility, and liquidity in the setting of cannabis industry stocks. Extending the three-factor Fama-French model, returns on cannabis stocks and liquidity are statistically positively correlated. Augmented investor interest increases returns. On the U.S. stock exchange, another study by Poutachidou and Papadamou (2021) concluded that there is a positive correlation between returns on stock market indices and increased investor attention on the U.S. Quantitative Easing policy, as measured by Google Metrics, which suggests that investor attention on QE seems to reduce volatility in the stock market and increase stock returns.

With that, there are gaps in academic research, as only few papers investigate the effectiveness of the application of machine learning approaches to not only forecast stock performance but also automate portfolio selection for higher returns and better investment decisions. Moreover, selecting the "right" securities to invest requires specific skill sets and experience as a number of criteria must be examined and a large amount of data must be analyzed while selecting securities. Hence, our focus is to leverage the use of machine learning techniques, namely Logistic Regression, Random Forest, and Extreme Gradient Boosting (XGBoost), in selecting investment securities based on three datasets: (1) technical indicators, (2) Google Trends search terms, and (3) the combination of the aforementioned. We sought to (1) study and identify the relevant factors in selecting securities for the investment portfolio, (2) develop a portfolio formation approach for individual investors, (3) evaluate the predictive capabilities of stock movement forecasting models using key-

words proxied from the three datasets above, and (4) compare the model performance between Logistic Regression, Random Forest, and Extreme Gradient Boosting (XGBoost).

The selected technical indicators indicate three characteristics. First is trend direction, namely (1) Simple Moving Average (SMA) used under normal market conditions, (2) Weighted Moving Average (WMA) used under unusual volatile market conditions, (3) Exponential Moving Average (EMA) suitable for analysis that need to prioritize the latest data. Second is reliability, namely (1) Moving Average Convergence Divergence (MACD), which provides information on both trend and momentum of the stock prices, and (2) Relative Strength Index (RSI), used to determine the probability of a trend reversal. Third is momentum, represented by Stochastic Oscillators K and D to compare the closing price with a range of prices for a given period of time (Bhargavi et al. 2017; Bustos and Pomares-Quimbaya 2020; Perry 2011; Praekhaow 2010; Vaidya 2018).

The Google Trends search dataset determines the popularity of keywords online such as product names, personal names, or possibly website names. Google Trends is created for users who rely on or benefit from trends, whether they are marketers or owners of online stores or even those who wish to start their own blog or vlog but are unsure of what to sell or how to determine the most frequently searched terms each day or month. Additionally, it aids in the development of marketing strategies and the gradual development of a business plan. In finance, Huang et al. (2019) discovered that the numerous directional movements of the S&P 500 vary in search volume based on the individual phrases searched for and, by extension, the sense of the word.

For forecasting models, the three machine learning models are applied in this study: Logistic Regression, Random Forest, and Extreme Gradient Boost (XGBoost). These approaches are used as comparison models to evaluate the prediction accuracy between the mentioned three datasets. The key measures used to compare the performance of the models include Receiver Operating Characteristic curves (ROC curves) and annualized and crisis backtest financial evaluations.

This study consists of (1) an introduction to explain the background and rationale of this study, (2) the methodology of research discussing the process and relevant theories used in this study, (3) the results and discussion to show the reflection from forecasting model performance and evaluations, and ends with (4) conclusions, limitations, and future workflow.

## 2. Methodology

Based on a machine learning algorithm, we constructed a generic common system, which aims to compare the effectiveness of stock forecasting models proxied by technical indicators, keywords from Google Trends, and a combination of both. According to Figure 1, this system is divided into four modules: dataset, data preprocessing, modelling, and evaluation. The details are as follows.

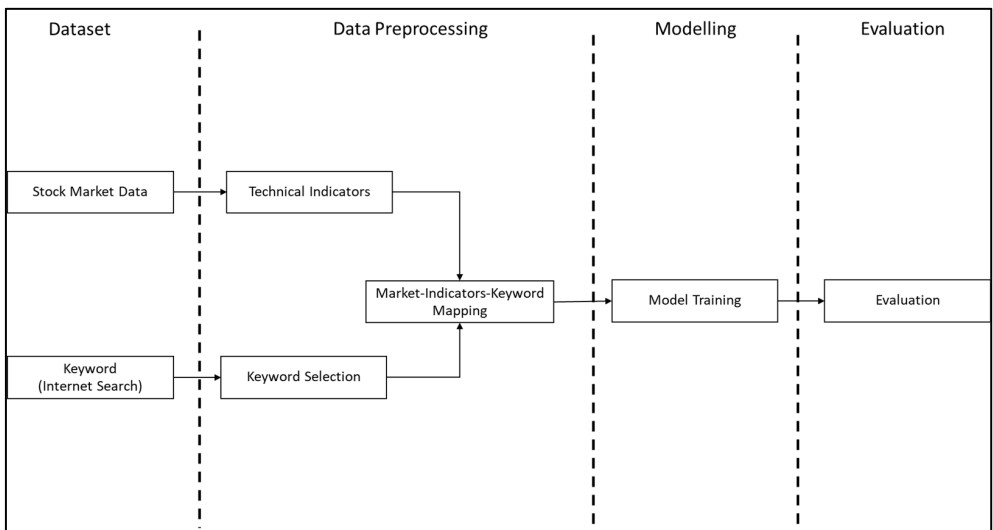

**Figure 1.** Generic common system components diagram.

### 2.1. Study Design

We would like to clarify that this study examines the price movement of the top 100 indices of the Stock Exchange of Thailand (SET100). WHA corporation is the example index in which we randomly pick from SET100.

For the stock price prediction, this study focuses on (1) the stock market data or technical indicators and (2) related keywords based on internet search.

As described above, the datasets come from two sources in the Thai stock market from 1 January 2017 to 31 December 2021:

1.  Yahoo Finance (stock market data)—weekly stock prices and trading volumes for stocks in SET100.
2.  Google Trends (keywords: internet search)—weekly search data for 484 specific terms which are commonly mentioned by the public on the internet. The selection of internet search terms are also based on the research by Preis et al. (2013) and the websites of Finnomena, SET, Krungsri, Encyclopedia, and stock2morrow.

Then, we processed the mentioned data to create technical indicators and ranked the top 20 keywords per each index which have both positive and negative correlations. The results from this data processing were used in the model creation.

### 2.2. Dataset

The data used in this study are structured data types, which are organized in a tabular format with definable columns. This layout enables easy access to information and simple or uncomplicated searching, without requiring additional work. We collected the data from two sources: the stock market and specific keyword searches.

### 2.2.1. Stock Market Data

Regarding stock market data, this information is public information available for download from the Stock Exchange's website, which includes the following:

*   The opening price is the first price of any listed stock at the start of a trading day.
*   The high and low values represent the stock's highest and lowest prices on that particular day. Generally, traders utilize these statistics to determine the volatility of a stock.
*   The closing price is the price of the stock at the close of the trading day.
*   The adjusted close price is regarded as the genuine price of that stock, as it reflects the stock's worth after dividends are distributed.

Stock prices are influenced by many factors and are often considered as one of the indicators to study market behaviors. As a result, using securities prices to study technical indicators improves the efficiency through which we can comprehend market behavior. The Stock Exchange of Thailand (SET100) data are collected weekly from Yahoo Finance. Examples of the daily historical stock prices and volumes collected from SET100 are shown in Table 1 for "WHA Corporation Public Company Limited" (WHA) stock.

**Table 1.** Examples of securities information of WHA Corporation Public Company Limited (WHA).

| Date | High | Low | Open | Close | Volume | Adj Close | Symbol |
|---|---|---|---|---|---|---|---|
| 4 January 2017 | 3 | 2.96 | 2.96 | 2.98 | 25,413,400 | 2.378223 | WHA |
| 5 January 2017 | 3.04 | 2.98 | 3 | 3.02 | 82,795,000 | 2.410144 | WHA |
| 6 January 2017 | 3.04 | 3 | 3.02 | 3.02 | 48,678,800 | 2.410144 | WHA |
| 9 January 2017 | 3.1 | 3.02 | 3.02 | 3.08 | $1.52 \times 10^8$ | 2.458028 | WHA |
| 10 January 2017 | 3.12 | 3.04 | 3.1 | 3.04 | 90,063,300 | 2.426106 | WHA |
| 11 January 2017 | 3.1 | 3.06 | 3.08 | 3.06 | 74,300,900 | 2.442067 | WHA |
| 12 January 2017 | 3.2 | 3.1 | 3.1 | 3.16 | $3.55 \times 10^8$ | 2.521873 | WHA |
| 16 January 2017 | 3.26 | 3.18 | 3.26 | 3.24 | 90,585,100 | 2.585719 | WHA |
| 17 January 2017 | 3.26 | 3.18 | 3.24 | 3.2 | 83,648,500 | 2.553796 | WHA |
| 18 January 2017 | 3.24 | 3.18 | 3.22 | 3.2 | 53,264,600 | 2.553796 | WHA |

2.2.2. Keywords (Internet Search)

The term "keywords" in this research refers to the terms used by investors to communicate in the securities investing sector, as well as the terms used by internet users to search for information on securities. This study consists of 484 keywords. There are eight categories according to research (Preis et al. 2013) and sources such as the Finnomena, SET, Krungsri, Encyclopedia website, and stock2morrow; these are basic investment terms, industry groups, stock names, trading methods, global search, popular words, idiom, and yearly search terms. The examples and definitions of keywords are exhibited in Table 2.

**Table 2.** Keyword examples.

| Keywords | | |
|---|---|---|
| **Type** | **Keyword** | **Definition** |
| Basic Investment Term | P/E | Price-to-Earnings (P/E) Ratio: The ratio for valuing a company that measures its current share price relative to its earnings per share (EPS) |
| Basic Investment Term | P/BV | Price to Book Value Ratio (P/BV): The market's valuation of a company relative to its book value |
| Basic Investment Term | EPS | Earnings Per Share (EPS): Calculated as a company's profit divided by the outstanding shares of its common stock |
| Industry Group | Agribusiness | Agribusiness |
| Industry Group | Food and Beverage | Food and beverage |
| Industry Group | Insurance | Insurance |
| Stock Name | ADVANC | Advanced Info Service PCL (ADVANC.BKK) |
| Stock Name | BBL | Bangkok Bank PCL (BBL.BKK) |
| Stock Name | CPN | Central Pattana PCL (CPN.BKK) |
| Trading Method | Technical | A trading strategy that primarily relies on technical indicators |

**Table 2.** *Cont.*

| Keywords | | |
|---|---|---|
| **Type** | **Keyword** | **Definition** |
| Trading Method | Day Trade | A trading strategy that is often informed by technical analysis of price movements and requires a high degree of self-discipline and objectivity |
| Trading Method | Swing Trade | A trading strategy that focuses on taking smaller gains in short term trends and cutting losses quicker |
| Global Search | Economics | Economics |
| Global Search | Politics | Politics |
| Global Search | Conflict | Conflict |
| Popular Word | กอง (Kong) | Mutual fund |
| Popular Word | ปอบ (Pop) | Broker |
| Popular Word | หรั่ง (Rang) | Foreign investor |
| Idiom | ลำไย (Lamyai) | Profit |
| Idiom | ซื้อควาย (Sue Khwai) | Buy stock(s) right before the stock's price goes down |
| Idiom | ขายหมู (Khai Mu) | Sale stock(s) right before the stock's price moves up |
| Yearly Search Term | คนละครึ่ง (Khon La Khrueng) | Thailand's government COVID-19 financial relief campaign |
| Yearly Search Term | โควิด-19 (COVID-19) | An infectious disease caused by the SARS-CoV-2 virus |
| Yearly Search Term | ชิมช้อปใช้ (Chim Chop Chai) | Thailand's government COVID-19 financial relief campaign |

Google Trends is a website of Google that was created to help determine the popularity of keywords online such as product names, personal names, or possibly website names. Users can view the popularity of these keywords by place, whether global, national, or provincial, and can also view daily popular trends (Huang et al. 2019; Nishimura and Acoba 2022; Papadamou et al. 2022; Poutachidou and Papadamou 2021; Sycinska-Dziarnowska et al. 2022; Tudor 2022). It is designed to even examine the keyword's popularity over time to determine potential development prospects. Google Trends is created for users who rely on or benefit from trends, whether they are marketers or owners of online stores or even those who wish to start their own blog or vlog but are unsure of what to sell, or how to determine the most frequently searched terms each day or month. Additionally, it aids in the development of marketing strategies and the gradual development of a business plan. In finance, Huang et al. (2019) discovered that the numerous directional movements of the S&P 500 vary in search volume based on the individual phrases searched for and, by extension, the sense of the word.

Google Trends normalizes its search data in order to make word comparisons easier. The following process normalizes the search results according to the time and location of the search query. Each data point is split by the total number of search locations and the time period over which the relative popularity is being compared. Otherwise, the location with the greatest amount of traffic is always ranked first. The result number is then scaled from 0 to 100 as a percentage of all searches across topics, areas exhibiting the same search

interest for a term (Huang et al. 2019; Nishimura and Acoba 2022; Papadamou et al. 2022; Poutachidou and Papadamou 2021; Sycinska-Dziarnowska et al. 2022; Tudor 2022). Table 3 illustrates the examples of popular words that the public used to search on Google ranked in Google Trends during 2017–2021 in Thailand.

**Table 3.** Google Trends search terms for each word in Thailand.

| Date | Debt | Color | Stocks | Restaurant | Portfolio | Inflation | Housing | Dow Jones |
|------|------|-------|--------|------------|-----------|-----------|---------|-----------|
| 1 January 2017 | 3 | 21 | 17 | 60 | 30 | 17 | 11 | 3 |
| 8 January 2017 | 22 | 31 | 32 | 51 | 29 | 24 | 7 | 11 |
| 15 January 2017 | 9 | 39 | 25 | 43 | 36 | 15 | 9 | 6 |
| 22 January 2017 | 37 | 41 | 42 | 43 | 38 | 24 | 32 | 9 |
| 29 January 2017 | 12 | 29 | 20 | 50 | 41 | 38 | 17 | 8 |
| 5 February 2017 | 34 | 38 | 21 | 51 | 25 | 55 | 30 | 8 |

### 2.3. Data Preprocessing

The data processing is divided into two sections: (1) technical indicators and (2) keyword selection. The technical indicators in the stock market refer to mathematical formulas based on the changes and direction of the market, such as prices and volumes. For keyword selection, the authors chose to study the search terms used by investors in the stock market, as well as the terms used by internet users to search for relevant information on securities.

### 2.3.1. Technical Indicators

In general, technical indicators are studied and plotted on charts to assist in providing directional information about a security's price. When the indices are used on a graph, they are displayed as lines, and the values are displayed between those lines alongside the price in order to determine the next likely direction of movement.

Technical indicators can be used to infer the behavior or trends of a time series and can be used to forecast the price of securities (Alfonso and Ramirez 2020; Anghel 2015; Basak et al. 2019; Bhargavi et al. 2017; Bustos and Pomares-Quimbaya 2020; Dash and Dash 2016; Perry 2011; Praekhaow 2010; Vaidya 2018). This study demonstrates how techniques can perform a summary rather than the full time series of securities prices, which simplifies machine learning. The technical indicators used in the forecasting model are as shown in Table 4.

**Table 4.** Technical indicators and formulas.

| Technical Indicators | Formula |
|----------------------|---------|
| Simple Moving Average (SMA) | $\frac{C_t + C_{t-1} + C_{t-2} + \ldots + C_{t-n-1}}{n}$ |
| Weighted Moving Average (WMA) | $\frac{(n)(C_t) + (n-1)(C_{t-1}) + \ldots + C_{t-n-1}}{(n) + (n-1) + \ldots + 1}$ |
| Exponential Moving Average (EMA) | $\alpha C_t + (1-\alpha)C_{t-1} + (1-\alpha)^2 C_{t-2} + \ldots + (1-\alpha)^{t-n+2} C_{t-n+2} + (1-\alpha)^{t-n+1} EMA_{t-n+1}$ |
| Moving Average Convergence Divergence (MACD) | $EMA12_t - EMA26_t$ |
| Relative Strength Index (RSI) | $100 - \frac{100}{1 + \left(\sum_{i=0}^{n-1} UP_{t-i}/n\right) \Big/ \left(\sum_{i=0}^{n-1} DW_{t-i}/n\right)}$ |
| Stochastic Oscillator K (K) | $\frac{C_t - L_{t-n}}{H_{t-n} - L_{t-n}} \times 100$ |
| Stochastic Oscillator D (D) | $\frac{\sum_{t=0}^{n=1} K_t}{n}$ |

Note: $C_t$ is the closing price, $L_t$ is the low price, $H_t$ is the high price at time t, $\alpha$ is a smoothing factor, $L_t$ and $H_t$ mean lowest low and highest high in the last t days, respectively. $UP_t$ means upward price change, while $DW_t$ is the downward price change at time t.

**The Moving Average (MA)** is a technical indicator that indicates the average trend over a specified time period by smoothing and filtering various abnormal signals and moving averages formed by using average closing prices over a specified time period.

**Simple Moving Average (SMA)**

This mean is appropriate in normal-volatility situations in a time series to help data analysts better interpret the patterns and contexts of stock market conditions (Perry 2011).

**Weighted Moving Average (WMA)**

This mean is appropriate in high-volatility situations in a time series to help data analysts better interpret the patterns and contexts of historical trends (Perry 2011).

**Exponential Moving Average (EMA)**

This average prioritizes the latest data. Therefore, it responds to price changes faster than a simple moving average (Bustos and Pomares-Quimbaya 2020; Praekhaow 2010).

**Moving Average Convergence Divergence (MACD)**

MACD indicates a trend that has the idea of drawing two moving averages at the same time and then analyzing the nature of the two moving averages. The special interesting point of MACD is that it is an indicator that can provide two pieces of information: the simultaneous view is the trend direction of the stock price (trend) and the momentum of the share price (momentum). The standard MACD is the 12-period EMA subtracted by the 26-period EMA (Alfonso and Ramirez 2020; Anghel 2015; Bustos and Pomares-Quimbaya 2020; Dash and Dash 2016).

**The Relative Strength Index (RSI)**

The Relative Strength Index (RSI) is a technical indicator that indicates whether an asset is strong or weak in relation to its recent closing price. Additionally, it is used to determine the probability of a trend reversal (Bhargavi et al. 2017).

**The Stochastic Oscillator**

The Stochastic Oscillator is a momentum technical indicator that is used to compare closing prices over a specified time period to a range of prices. This oscillator is extremely sensitive to market price changes. The indicator's volatility can be smoothed somewhat by changing the time interval being measured. The most frequently used stochastic oscillators are Stochastic Oscillator K (K) and Stochastic Oscillator D (D) (Vaidya 2018). The K line compares the lowest low and the highest high of a given period to define a price range, then displays the last closing price as a percentage of this range. The D line is a moving average of K.

2.3.2. Keyword Selection

This is the selection of factors that affect each security. The factors mentioned are internet search keywords and the Pearson Correlation Coefficient (r) in Equation (1). The top 20 keywords, both positive and negative correlations, are selected for each security. The top 20 keywords, both positive and negative, of WHA Corporation Public Company Limited are shown in Table 5.

$$r_{xy} = \frac{n(\sum xy) - (\sum x)(\sum y)}{\sqrt{[n\sum x^2 - (\sum x)^2][n\sum y^2 - (\sum y)^2]}} \tag{1}$$

where

$r_{xy}$: Pearson Correlation Coefficient between variables x and y.
$\sum x$: The sum of the measured data from variable x.
$\sum y$: The sum of the measured data from variable y.
$\sum xy$: The sum of product of the variables x and y.
$\sum x^2$: The sum of the squares of variable x.
$\sum y^2$: The sum of the squares of variable y.
n: Number of data.

**Table 5.** Top 20 keywords of WHA Corporation Public Company Limited.

| Symbol | Positive | Correlation | Negative | Correlation |
|---|---|---|---|---|
| WHA | wha | 0.2322 | restaurant | −0.2028 |
| WHA | sta | 0.2262 | major | −0.1999 |
| WHA | tcap | 0.2161 | holiday | −0.1818 |
| WHA | sawad | 0.2066 | fun | −0.1771 |
| WHA | settrade | 0.1885 | แขก (kaek) | −0.1693 |
| WHA | dow jones | 0.1878 | hybride | −0.1671 |
| WHA | banpu | 0.1851 | food& beverage | −0.1642 |
| WHA | tisco | 0.1840 | bec | −0.1567 |
| WHA | toa | 0.1832 | ประชัย (bpra chai) | −0.1511 |
| WHA | aot | 0.1824 | forex | −0.1430 |
| WHA | bcp | 0.1822 | short selling | −0.1411 |
| WHA | hmpro | 0.1766 | water | −0.1362 |
| WHA | ปอด (bpot) | 0.1753 | mbk | −0.1340 |
| WHA | lh | 0.1724 | thani | −0.1339 |
| WHA | amata | 0.1712 | เบาะหนัง (bor nang) | −0.1331 |
| WHA | bdms | 0.1689 | travel | −0.1249 |
| WHA | cpf | 0.1679 | top | −0.1224 |
| WHA | bbl | 0.1672 | ตกรถ (dtok rot) | −0.1218 |
| WHA | บริการ (bor-ri-gaan) | 0.1645 | markets | −0.1214 |
| WHA | cpn | 0.1615 | bts | −0.1198 |

Note: แขก (kaek): Indorama Ventures PCL (IVL) stock, ประชัย (bpra chai): TPI Polene PCL (TPIPL) stock, เบาะหนัง (bor nang): Interhides PCL (IHL) stock, ตกรถ (dtok rot): about to buy the stock(s) when the price is considerably low but then it goes up, ปอด (bpot): stock portfolio, บริการ (bor-ri-gaan): industry group service.

### 2.4. Modeling

Modeling is the relationship of data in various forms. Modeling in this study includes the Classification Model, which is a supervised learning model to create a model that must have variables used to measure the target as a starting point to learn. The goals of the classification will be grouped or discrete such as yes/no, A/B/C, etc. Therefore, in evaluating the results obtained from the classification model, accuracy can be measured, e.g., with a confusion matrix.

The following types of modeling were used to predict the selection of securities in this study. We chose Logistic Regression, Random Forest, and XGBoost because (1) the three models were commonly used and cited in many relevant studies (Ananthakumar and Sarkar 2017; Basak et al. 2019; Geurts et al. 2006; Ghatasheh 2014; Sadorsky 2021) and (2) they can manage the imbalance data as also discussed (Le et al. 2022; Wang et al. 2022).

### 2.4.1. Logistic Regression

Logistic Regression is classified as a supervised learning model that is used to predict the probability of occurrence of an event of interest from a dataset of appropriate independent variables and algorithms based on mathematical equations. If breaking a Logistic Regression follows the type of independent variables, it can be divided into two types: Simple Logistic Regression, which has only one independent variable, and Multiple Logistic Regression, which has many independent variables. It is widely used in applications such as predicting the likelihood of a customer incurring bad debt to a banking company or predicting the likelihood of a customer migrating to a telephone network. This technique has also been used in applied marketing to discover market segmentation in order to maximize the chance of campaign offers for each market category (Ananthakumar and Sarkar 2017; Robles et al. 2008). Figure 2 depicts the approach for developing a Logistic Regression model, with further information provided below.

- Use a dataset to create a simple linear regression or multiple linear regression depending on the independent variables used in the type of work performed.
- Bring the regression equation to the Sigmoid function to adjust the value to be in the range 0–1 because the regression equation can have values greater than 1 or less than 0. It should be between 0 and 1 only, so this function has been implemented.
- By passing the sigmoid function, the probability of the event of interest is obtained.

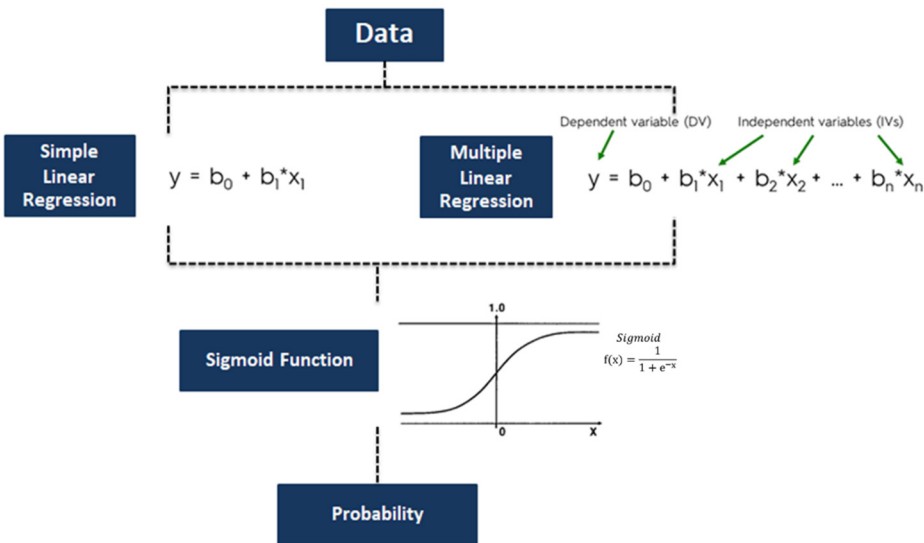

**Figure 2.** Logistic Regression modeling process diagram.

2.4.2. Random Forest

Random Forest is a tree-based model composed of several decision trees. It is a predictive model based on the Wisdom of the Crowd technique. This is also referred to as the Ensemble Technique and it is a type of Bootstrap Aggregation that is used to sample data and then build a predictive model. In Random Forest, the random sampling with replacement method is used so that the size of the new dataset is the same as the original dataset. This modeling provides high accuracy and low variance, which is better than a single decision tree but takes more complex calculations (Basak et al. 2019; Sadorsky 2021). There is one study that applied this modeling to predict credit risk prediction and compared it with more than 10 other models (Ghatasheh 2014). The study found that Random Forest generated the highest accuracy, sensitivity, and F-Measure. The main modeling processes are shown in Figure 3 and are described below:

- In Random Forest, n number of random records are taken from the dataset having k number of records (Bootstrapped Dataset).
- Individual decision trees are constructed for each sample.
- Each decision tree will generate an output.
- The final output is considered based on majority voting.

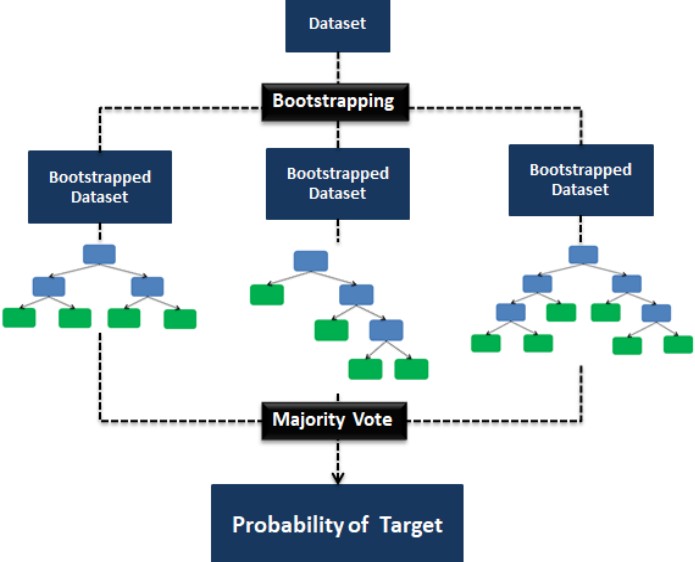

**Figure 3.** Random Forest modeling process diagram.

### 2.4.3. Extreme Gradient Boosting (XGBoost)

Extreme Gradient Boosting (XGBoost) is a model classified as "Boosting". It uses a low-precision weak classifier to predict data and then uses the resulting weak classifier to correct the error of the first created model. The predicted sum is a new classification that is repeated over and over until the best model is obtained from the prediction sum. This model employs the same ensemble technique as the Random Forest classification model. However, the process of creation is clearly distinct. The strength of this model is that, in comparison to other supervised models, it is capable of predicting data with extremely complex and accurate data. It is widely used in both regression and classification problems. However, the computation and processing take a long time because the model is a hierarchical operation, unlike Bootstrap Aggregation, which can run concurrently (Basak et al. 2019; Geurts et al. 2006). The steps for creating the master model are shown in Figure 4, and the details are as follows.

Step 1: Identify the default criteria used for forecasting. The criteria are classified by (1) probabilities: probabilities greater than or equal to 0.5 will be forecasted as "positive class", and probabilities less than 0.5 but greater than 0 will be forecasted as "negative class"; and by (2) types of targets.

Step 2: Calculate the residual from the actual conversion of the forecasted variable by setting the positive class to 1 and negative class to 0. Then, subtract the true value of the variable from the criteria obtained in Step 1.

Step 3: Create the decision tree from all independent variables; this decision tree is used to predict the residual from Step 2.

Step 4: Calculate the residual sum of all leaf nodes from the formula below.

$$\text{Residual}_{\text{New}} = \frac{\sum \text{Residual}}{\sum \left(\text{Previous Probability} \times (1 - \text{Previous Probability})\right) + \lambda}$$

Step 5: Combine the criteria obtained in Step 1 through the log odds function with the threshold values created in Step 3 via the decision tree and utilize the learning rates as a parameter to help tune the decision tree so that it does not overfit the data. When the learning rates are between 0 and 1, we obtain a weak classifier.

Step 6: Recalculate the residual value from the weak classifier in Step 5 with all the data calculated. Then, take the value of each data point through the Sigmoid function, which will derive the probability of each of the data.

Step 7: Repeat Step 2 until the residual value is unchanged or reaches the specified number of decision trees.

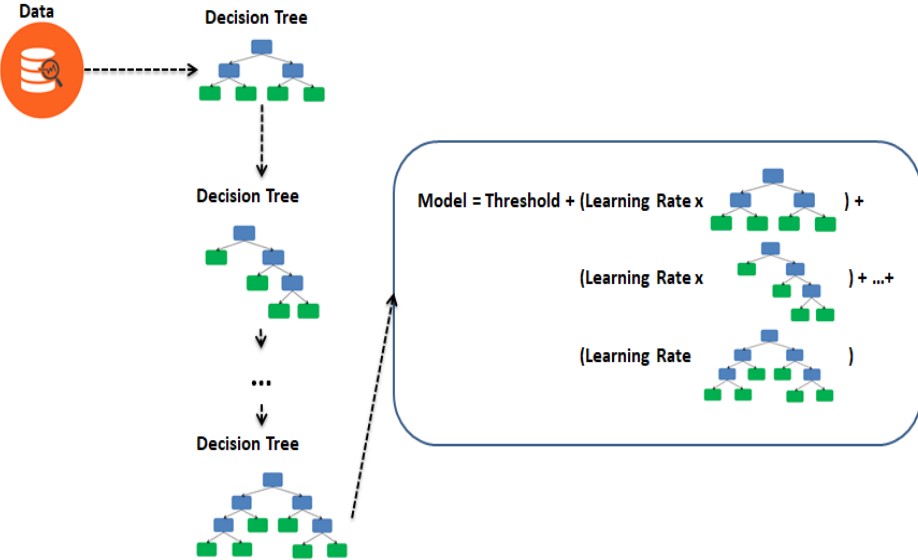

**Figure 4.** Extreme Gradient Boosting (XGBoost) modeling process diagram.

*2.5. Evaluation*

Evaluation is an important tool for measuring the effectiveness of this research. The tools used in the evaluation must be consistent with the measurements of performance. In this study, we opted to use the Receiver Operating Characteristic curves as follows.

**Receiver Operating Characteristic curves (ROC curves)**

ROC curves are frequently used to assess the effectiveness of binary classification algorithms. They display the performance of a classifier graphically, rather than as a single value, as most other metrics do (Ananthakumar and Sarkar 2017; Basak et al. 2019; Bustos and Pomares-Quimbaya 2020; Huang et al. 2019; Trifonova et al. 2014). The dependent variable (Y) in this case is a qualitative variable, which can be divided into two cases: $Y = 1$ when the interesting event or test result is positive, and $Y = 0$ when the incident is ignored, or the test result is negative.

The cut-off point is the point used to classify events into interesting and uninteresting events. Comparisons between the forecast values and observations can be divided into 4 cases as shown in Table 6.

**Table 6.** Confusion matrix.

|  | Actually Positive (1) | Actually Negative (0) |
|---|---|---|
| Predicted Positive (1) | True Positive (TP) | False Positive (FP) |
| Predicted Negative (0) | False Negative (FN) | True Negative (TN) |

TP, TN, FP, and FN in the table is represented by the frequency values:

- A True Positive (TP) response is when what is predicted matches what is actually happening. In the case of a prediction that is "true", what happened is "true".
- A True Negative answer (TN) is when what the prediction matches what happened. In the event that the prediction is "not true", what happened is "not true".
- A False Positive (FP) is a prediction that does not match what happened, that is, a prediction is "true", but what happens is "not true".
- False Negative (FN) is a prediction that does not match what actually happened, which is a prediction that something is "not true", but what happens is "true".

The True Positive Rate (TPR) is a measure of when what is predicted matches what is actually happening with all true events (True Positive (TP) + False Negative (FN)), as shown in Equation (2).

$$\text{TPR} = \frac{\text{TP}}{\text{TP} + \text{FN}} \qquad (2)$$

The False Positive Rate (FPR) is a measure of when what is predicted does not match what is happening with all false events (True Negative (TN) + False Positive (FP)), as shown in Equation (3).

$$\text{FPR} = \frac{\text{FP}}{\text{TN} + \text{FP}} \qquad (3)$$

Precision is the weighted average of positive and negative precision, while recall is the weighted average of positive recall, as shown in Equations (4)–(7).

$$\text{Precision}_{\text{positive}} = \frac{\text{TP}}{\text{TP} + \text{FP}} \qquad (4)$$

$$\text{Precision}_{\text{negative}} = \frac{\text{TN}}{\text{TN} + \text{FN}} \qquad (5)$$

$$\text{Recall}_{\text{positive}} = \frac{\text{TP}}{\text{TP} + \text{FN}} \qquad (6)$$

$$\text{Recall}_{\text{negative}} = \frac{\text{TN}}{\text{TN} + \text{FP}} \qquad (7)$$

Accuracy and F1-score are estimated using Equations (8) and (9).

$$\text{Accuracy} = \frac{TP + TN}{TP + FP + TN + FN} \tag{8}$$

$$\text{F1-score} = 2 \times \frac{\text{Precision} \times \text{Recall}}{\text{Precision} + \text{Recall}} \tag{9}$$

ROC curves are plotted between TPR (Sensitivity) and FPR (1-Specificity) for all possible cut-off values. The y-axis represents TPR, while the x-axis represents FPR. At each cut-off value, it separates the forecast results into two groups: the event where P(Y = 1) is greater than or equal to the cut-off point, and the event where P(Y = 1) is less than the cut-off point, where P(Y = 1) is the probability that the interesting event or test result is positive. The area under the ROC curve is an index used to indicate the accuracy or reliability of a model. The model with the greatest area under the ROC curve is the most accurate model (Elizabeth et al. 1988)

## 3. Results and Discussion

### 3.1. Performance on Dataset

The comparative data are grouped into three categories: (1) Technical Indicators, including SMA, WMA, EMA, MACD, RSI, K, and D; (2) Google Trends keywords; and (3) Technical Indicators and Keywords. Additionally, machine learning data are divided into three sets: Train and Test, which is 1 January 2017 through 30 May 2021 with an 80:20 ratio, and Unknown, which is 1 June 2021 to 31 December 2021. The unknown dataset is used for the purpose of knowing the performance on unseen data.

Table 7 shows that during the data test process, the combination of Keywords and Indicators outperforms other datasets in terms of accuracy, precision, recall, F1-score, and ROC-AUC score. For unknown data, Random Forest and XGBoost models with Keywords and Indicators also outperform other datasets. Specifically, Keywords and Indicators coupled with Logistic Regression perform better when comparing F1-score and ROC-AUC scores.

**Table 7.** Performance on dataset.

| Model | Type Dataset | Test | | | | | Unknown | | | | |
|---|---|---|---|---|---|---|---|---|---|---|---|
| | | Accuracy | Precision | Recall | F1-Score | AUC | Accuracy | Precision | Recall | F1-Score | AUC |
| Logistic Regression | Indicators | 0.9699 | 0.9589 | 0.9699 | 0.9587 | 0.8457 | 0.9818 | 0.9739 | 0.9818 | 0.9745 | 0.8251 |
| | Keywords | 0.9727 | 0.9664 | 0.9727 | 0.9653 | 0.8999 | 0.9840 | 0.9843 | 0.9840 | 0.9779 | 0.8880 |
| | Keywords and Indicators | 0.9813 | 0.9794 | 0.9813 | 0.9790 | 0.9767 | 0.9834 | 0.9827 | 0.9834 | 0.9830 | 0.9733 |
| Random Forest | Indicators | 0.9694 | 0.9555 | 0.9694 | 0.9553 | 0.9348 | 0.9818 | 0.9638 | 0.9818 | 0.9727 | 0.9324 |
| | Keywords | 0.9727 | 0.9734 | 0.9727 | 0.9619 | 0.9120 | 0.9814 | 0.9717 | 0.9814 | 0.9737 | 0.9045 |
| | Keywords and Indicators | 0.9734 | 0.9708 | 0.9734 | 0.9642 | 0.9645 | 0.9831 | 0.9833 | 0.9831 | 0.9758 | 0.9623 |
| XGBoost | Indicators | 0.9732 | 0.9674 | 0.9732 | 0.9681 | 0.9577 | 0.9818 | 0.9767 | 0.9818 | 0.9782 | 0.9499 |
| | Keywords | 0.9743 | 0.9694 | 0.9743 | 0.9682 | 0.9339 | 0.9850 | 0.9840 | 0.9850 | 0.9802 | 0.9456 |
| | Keywords and Indicators | 0.9823 | 0.9810 | 0.9823 | 0.9796 | 0.9787 | 0.9879 | 0.9871 | 0.9879 | 0.9855 | 0.9810 |

Overall, when comparing the predictive performance of the three datasets, the Keywords and Indicators dataset has the highest F1-score and ROC-AUC score across the three models. We imply that the datasets can reveal online search trends and financial time series that can lead to stock analysis. Therefore, Table 8 uses the Keywords and Indicators dataset for further analysis.

**Table 8.** Performance for crisis.

| % Success | Model | | | Win |
|---|---|---|---|---|
| **Crisis** | **Logistic Regression** | **Random Forest** | **XGBoost** | **Model** |
| QE | 55.00% | 85.00% | 83.75% | Random Forest |
| Controversy | 51.11% | 78.89% | 65.56% | Random Forest |
| Foreign Investors | 45.00% | 71.67% | 63.33% | Random Forest |
| MSCI | 57.78% | 86.67% | 82.22% | Random Forest |
| COVID-19 (1) | 51.54% | 73.85% | 79.23% | XGBoost |
| Protestation | 65.56% | 94.44% | 90.00% | Random Forest |
| COVID-19 (2) | 42.86% | 90.00% | 91.43% | XGBoost |
| COVID-19 (3) | 40.00% | 82.22% | 84.44% | XGBoost |
| COVID-19 (4) | 56.25% | 92.50% | 93.75% | XGBoost |
| COVID-19 (5) | 51.11% | 90.00% | 90.00% | Random Forest |
| AVERAGE | 51.62% | 84.52% | 82.37% | Random Forest |

*3.2. Performance*

We conducted a stock market simulation based on weekly trading. For the strategy, we determined buying at the closing price of the previous week's final trading day and selling at the week's final day's closing price. When simulating in the Thai stock market, the first week and the investment in the following weeks will increase or decrease depending on the investment results of the previous week, where the model selects the top 10 most probable stocks. Additionally, we split trading for each year and every year beginning with 10,000 Thai baht from 2017 to 2021. Performance is determined by the percentage of successful stock picks (percentage of right weekly stock picks) and the percent annualized return (total return each week over 1 year). The results of Logistic Regression, Random Forest, and XGBoost are compared in Figures 5–8, which detail the performance of stock selection and total return on an annual basis from 2017 to 2021.

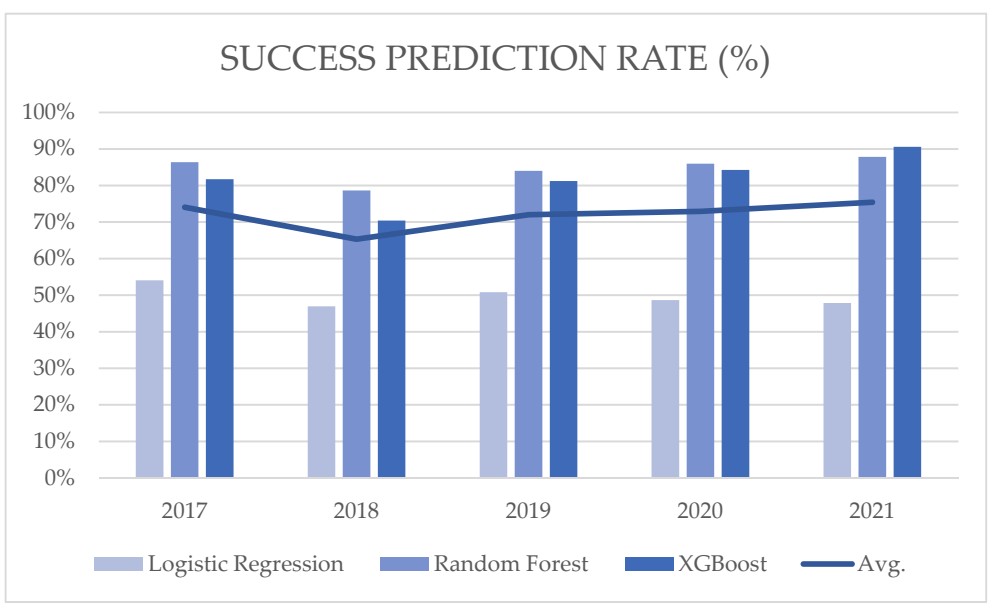

**Figure 5.** Stock selection performance (success prediction rate).

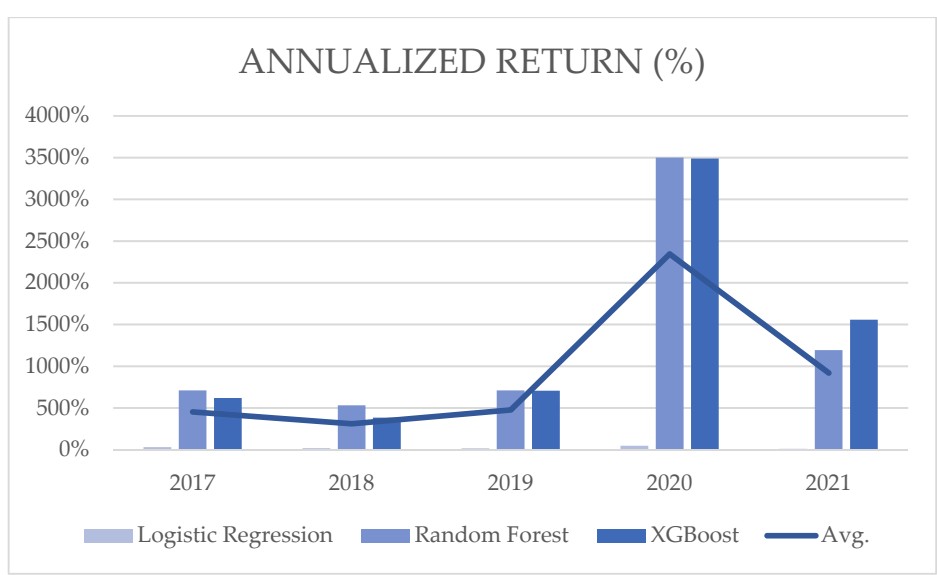

**Figure 6.** Stock selection performance (annualized return).

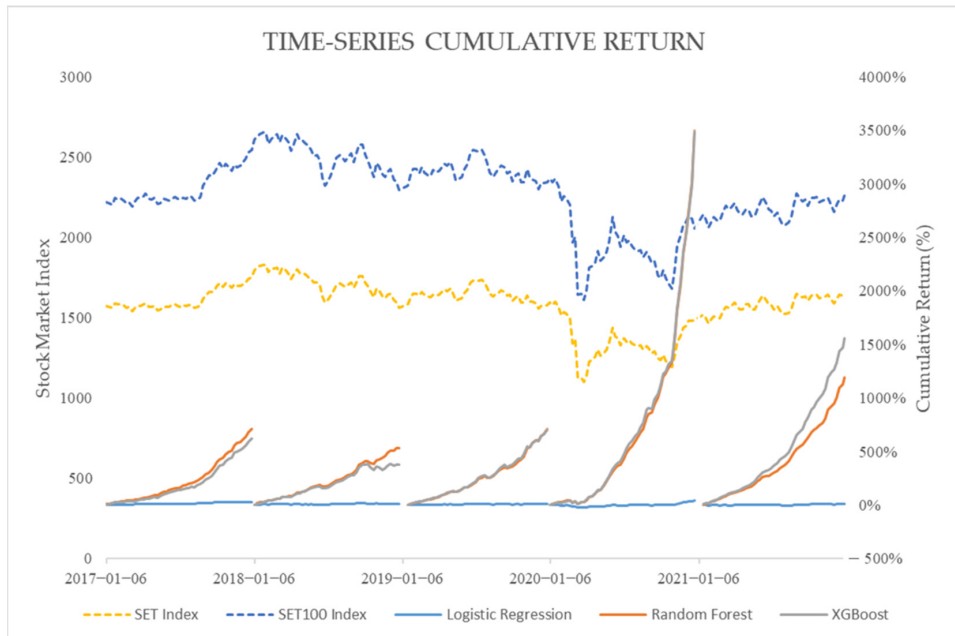

**Figure 7.** Time series cumulative return.

The following are the annual results from 2017 to 2021 (Figures 5 and 6). The average success rate for the Logistic Regression model is 49.65% and the average annualized return is 23.25%, with the highest success prediction rate in 2017 being 54.04% and the highest annualized return in 2020 being 45.45%. The Random Forest model has an average success rate of 84.56% and an average annualized return of 1328.43%, with the best success prediction rate in 2021 being 87.84% and the best annualized return in 2020 being 3500.66%. The XGBoost model has an average success rate of 81.63% and an average annualized return of 1350.38%, with a success rate of 90.59% for the best year 2021 and 3489.08% for the best year 2020. From Figure 7, the Logistic Regression, Random Forest, and XGBoost portfolios show dominance over the benchmarks SET index and SET100 index in the portfolio across the simulation periods (2017 to 2021). The dominance relationship becomes more significant starting during the COVID-19 outbreak in 2020. In conclusion, performance comparisons with the annualized backtest financial evaluation reveal the distinction between the three

models. Clearly, the use of Random Forest has the highest average percentage of success, whereas XGBoost has the highest average annualized return.

**Figure 8.** Crisis in Thailand during the years 2017–2021.

In 2020, it was the year with the most return because the Thai stock market was extremely volatile due to various COVID-19 events and crises. Every year, there are events that have an impact on the stock market, both positive and negative. Thus, Table 8 demonstrates the model's performance for several crises in Thailand.

### 3.3. Performance for Crisis

During the various crises in Thailand, the performance of models differs between periods and models. From 2017 to 2021, according to Figure 8, there are six types of crises: Quantitative Easing (QE), Controversy, Foreign Investors, MSCI, Protestation, and COVID-19. For COVID-19 crises, it can be classified into five states of COVID-19, as each state is pretty unique in terms of model performances in the Thai stock market. Hence, we explain the states of COVID-19 outbreaks and draw attention to their impacts during each phase. Table 8 displays the performance of stock selection during various times of crisis.

#### 3.3.1. Quantitative Easing (QE)

Quantitative Easing (QE) is a form of monetary policy borne by each country's central bank. In January 2018, the Stock Exchange of Thailand index closed at its all-time high of 1838.96 points, which was a significant factor in the decline of gold prices (Chai et al. 2021). This was due to the impact of QE measures in the United States and Japan and the reduction in interest rates by the European Central Bank (ECB), which encouraged investors to invest in stock markets in search of higher yields. The results of Logistic Regression, Random Forest, and XGBoost from January to February 2018, as shown in Table 8, revealed successful stock pick rates of 55.0%, 85.0%, and 83.75%, respectively.

#### 3.3.2. Controversy

The U.S. and China's trade retaliation is a global trade war. Particularly, trade countermeasures between the United States and China have caused the SET index to decline for the majority of the week, although there are factors supporting energy stocks due to the rise in oil prices on the global market. However, this is insufficient to offset the pressure

exerted by foreign investors' net selling of Thai stocks from the beginning to the middle of the week. However, the Thai stock market still closed in the red in contrast to regional stock markets. In June and July 2018, the successful stock selection rates for Logistic Regression, Random Forest, and XGBoost were 51.11%, 78.89%, and 65.55%, respectively, as shown in Table 8.

### 3.3.3. Foreign Investors

The SET market decreased dramatically to 1548.37 points, the lowest in 19 months on 27 December 2018. The major reason is that the foreign investors sold a net 1832.50 million baht (Smart 2018; Today 2018). As shown in Table 8, the results of Logistic Regression, Random Forest, and XGBoost for the period December 2018 to January 2019 showed stock selection performance rates of 45.00%, 71.67%, and 63.33%, respectively.

### 3.3.4. MSCI

MSCI Index is Morgan Stanley Capital International's benchmark. It was developed as a metric for foreign investors to use in selecting stocks and returns. On 28 May 2019, the Thai stock market closed at 1632.04 points, a gain of 7.20 points, with a trading value of 204,855.67 million baht, a record high. This was due to the reweighting of the MSCI Thailand index, which boosted the weight of some Thai equities in the index's investment. According to Table 8, the stock selection rates predicted by Logistic Regression, Random Forest, and XGBoost for the period of May–June 2019 were 57.78%, 86.67%, and 82.22%, respectively.

### 3.3.5. COVID-19 (1)

The COVID-19 crisis has had various effects on the global economy, with the lockdown measures having the most economic impact. When there are lockdowns, it makes it harder for people and businesses to buy things in public places such as department stores. It began affecting the Thailand Stock Exchange on 26 February 2020. The Thai Stock Exchange fell sharply by 72.69 points, or −5.05%, in a single day, closing at 1366.41 points. The SET ended trading on 9 March at 1255.94 points, which was down 108.63 points or −7.96%. The SET for 12 March closed at 1114.91 points, a decrease of 134.98 points, or a loss of 10.80%. The stock market closed on 16 March at 1046.08 points, with a loss of 82.83 points, or 7.34%. As indicated in Table 8, the performance of stock selection from February to April 2020 using Logistic Regression, Random Forest, and XGBoost were 51.54%, 73.85%, and 79.22%, respectively.

### 3.3.6. Protestation

Protests in Bangkok, Thailand, on 30 October 2020 led to the declaration of a state of emergency in the capital city. Because of this, the SET index closed the trading day at 1214.95 points. Using Logistic Regression, Random Forest, and XGBoost, the success prediction rates for October and November 2020 were as follows (Table 8): 65.56%, 94.44%, and 90%, respectively.

### 3.3.7. COVID-19 (2)

The COVID-19 pandemic began in the province of Samut Sakhon in Thailand at the same time that the country instituted quarantine measures to stop the virus from spreading further. With investors concerned about the closure of Samut Sakhon, the Stock Exchange of Thailand Index decreased by more than 5% on 21 December 2020. The market closed with a loss of 80.60 points, or −5.44%, at 1401.78 points. According to Table 8, the stock selection success rates predicted by Logistic Regression, Random Forest, and XGBoost for the period December 2020 to January 2021 were 42.86%, 90%, and 91.4%, respectively.

### 3.3.8. COVID-19 (3)

As a consequence of the relaxation of lockdown measures, the Stock Exchange of Thailand index ended on 11 June 2021 at 1636.56 points, a gain of 11.29 points, the highest level in 1 year, 4 months, and 18 days. However, with the easing of the lockdown measures, the epidemic became more severe, and the stock market continued to decline. According to Table 8, the performance of successful stock picks using Logistic Regression, Random Forest, and XGBoost for the period of June to July 2021 was 40%, 82.22%, and 84.44%, respectively.

### 3.3.9. COVID-19 (4)

As the number of COVID-19 cases in the country continued to exert more pressure, the Stock Exchange of Thailand closed at 1521.72 points on 6 August 2021, a fall of 5.94 points and the lowest close in five months. This number is still far over 20,000 every day. The market is well aware of the issue, yet it still poses risks. Investors are wary as a result. According to a tally compiled by the Stock Exchange of Thailand (SET) on 6 August 2021, institutional investors sold 135.90 million baht worth of shares, brokers sold 363.93 million baht worth of shares, foreign investors sold 2078.31 million baht worth of shares, and retail investors bought 2578.14 million baht worth of shares. Foreign investors' sales over the first six days of August totaled as much as 8148.43 million baht. As can be shown in Table 8, the success rates of stock picks made using Logistic Regression, Random Forest, and XGBoost during the months of August and September 2021 were 56.25%, 92.50%, and 93.75%, respectively.

### 3.3.10. COVID-19 (5)

On 28 November 2021, the Stock Exchange of Thailand dropped 37.85 points, closing at 1610.61 as a result of the Omicron strain of the 2019 coronavirus. Continuing its downward trend, the index shed another 20.92 points on 29 November 2021, ending the day at 1589.69. It dropped another 21.00 points on 30 November 2021, finishing at 1568.69. Table 8 displays that the accuracy rates of Logistic Regression (51.11%), Random Forest (90.00%), and XGBoost (90.00%) were the lowest and highest for predicting profitable stocks during November and December of 2021, respectively.

Based on the various crises that occurred in Thailand between 2017 and 2021, the crisis is divided into six categories: QE, Controversy, Foreign Investors, MSCI, COVID-19, and Protest. The overall performance of Logistic Regression is proxied by %success, which in this experiment was 51.62% on average. Random Forest has an overall accuracy rate of 84.52% (the highest performance during the financial crisis: QE, Controversy, Foreign Investors, MSCI, COVID-19 (5), and Protest). For XGBoost, the overall performance reflects 82.37% success (the highest performance during the crisis: COVID-19 (1–5)). Lastly, for backtest financial evaluation during the crisis, Random Forest ranked first in terms of %success, with 84.52%, while XGBoost ranked the first during COVID-19 outbreaks with 87.77%.

Based on the above performance, Random Forest had a higher stock selection rate than the other two models (Logistic Regression and XGBoost) because its algorithm was very stable, and it works well with both continuous and discrete variables. Even if a new data point is added to the dataset, the overall algorithm is not significantly impacted because it is unlikely that the new data will affect all trees (Breiman 2001). However, during COVID-19 when the market fell sharply, XGBoost had a higher stock selection rate. The stock market experienced a sharp decline during the COVID-19 period due to the nature of the data. It is possible that Random Forest will assign less weight to this class, but XGBoost is an excellent alternative for unbalanced datasets. Consequently, XGBoost has a higher stock selection rate, whereas Logistic Regression has a relatively low performance. Logistic Regression is optimized for discrete variables (Robles et al. 2008). Compared to Random Forest and XGBoost, the Keywords and Indicators dataset characteristics are complex, having both continuous and discrete variables data characteristics.

## 4. Conclusions

### 4.1. Conclusion

To address the challenges found in the existing literature on portfolio formation with stock selection, we aimed to (1) study and identify relevant factors in selecting securities for the investment portfolio, (2) develop a portfolio formation approach for individual investors, (3) evaluate the predictive capabilities of stock movement forecasting models using keywords proxied from the three datasets (technical indicators, Google Trends search terms, and the combination of the aforementioned), and (4) compare the model performance between Logistic Regression, Random Forest, and Extreme Gradient Boosting (XGBoost). With the use of the machine learning (ML) models and technical indicators combined with Google Trends search terms, the proposed prediction model was constructed, bringing the benefits of machine learning to stock selection. Several conclusions can be drawn from this research.

First, the predictive accuracy of Logistic Regression, Random Forest, and XGBoost models was initially evaluated using the three datasets mentioned above and compared. All three models have the same maximum ROC-AUC Score in the dataset, which combines technical indicators and Google Trends search terms. The combination of datasets can be applied to better understand the internet search trends as well as financial time series, which in turn can lead to more precise stock analysis.

Second, the three ML approaches were used as comparison models to assess the predictive performance with the key parameters: ROC curves, annualized backtests, and financial crisis backtests. For ROC curves, there was no clear distinction between the performances of the three ML models, while XGBoost performed the best. Backtest evaluation (annualized) made the contrast clearer, Random Forest had the highest average success rate, and XGBoost had the highest average annualized returns. In the crisis situation, Random Forest had the highest average success rate. During the COVID-19 crisis, XGBoost proved to have the highest average success rate. In summary, this study concluded that the backtest evaluation was more suitable to compare the performance of each ML method than ROC curves. The backtest evaluation was able to incorporate real events in the stock markets into the comparison model.

Finally, the results from the prediction procedure revealed that Random Forest and XGBoost were nearly identical but still different. During moderately volatile markets, the Random Forest model outperforms the XGBoost model in terms of both average success rate and average annualized return. Nevertheless, the XGBoost model performs significantly better during periods of extreme market volatility, such as the COVID-19 pandemic. The algorithm has the capability to identify stocks with a high future price growth potential. This model is superior to the Random Forest model in both aspects (average success rate and average annualized return).

### 4.2. Limitation and Future Work

Despite the fact that this study provides useful insights, it has some limitations. First, we only analyze Thai Stock data. Hence, the result from this study might not be generalizable for other economies and countries. Second, we utilized the Keywords and Indicators dataset to determine which keyword is appropriate for a specific time frame. Consequently, it is essential to update new keywords and ensure the correlation between stock movements and keywords in different time periods. Third, this study only used the Pearson Correlation Coefficient (r) to determine the relationship between stocks and keywords. In the future, it would be better to expand the study and perform a sentiment analysis determining the specific correlation of keywords that influence individual stocks. Fourth, there is a limit on the number of shares that may be imported to the portfolio with the same amount every time. This is because this research only chose companies with the best chance of price growth for a total portfolio of 10 stocks every weekly transaction per round. In fact, the number of shares that will give the best return in each round unnecessarily equals the number of equities in the portfolio. If optimization algorithms are used

to construct an optimal portfolio, the portfolio may perform better and be better overall. Finally, since this study did not examine the recommended buy/sell cut-off points for each stock each week, a suggested buy/sell cut-off point could increase profits more effectively.

All in all, future research should apply sentiment analysis for keywords and individual stocks, examine algorithm optimization for portfolio selection, and take into account the suggested buy/sell cut-off points for each week.

**Author Contributions:** Conceptualization, K.S. and J.Y.; Methodology, K.S. and J.Y.; Software, K.S.; Validation, K.S. and J.Y.; Formal analysis, K.S. and J.Y.; Investigation, J.Y.; Resources, K.S.; Data curation, K.S.; Writing—original draft preparation, K.S. and J.Y.; Writing—review and editing, J.Y.; Visualization, K.S.; Supervision, J.Y.; Project administration, J.Y. All authors have read and agreed to the published version of the manuscript.

**Funding:** This research received no external funding.

**Informed Consent Statement:** Not applicable.

**Data Availability Statement:** Not applicable.

**Acknowledgments:** The authors would like to acknowledge Praewa Kulatnam for her contribution in proofreading and editing this manuscript.

**Conflicts of Interest:** The authors declare no conflict of interest.

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
