# Peer review of "Stock Movement Prediction Using Machine Learning Based on Technical Indicators and Google Trend Searches in Thailand"

_ijfs, doi:10.3390/ijfs11010005_

Round 1

Reviewer 1 Report

In this paper, the authors suggested a stock movement prediction ML method using google trend and technical indicators. The following comments may be considered in the revised manuscript.

1.    Other performance-checking values should also be specified. For example,

(1) Mean Forecast Error (MFE)

(2) Mean Absolute Percentage Error (MAPE)

(3) Relative Absolute Error (RAE)

2.     The statistical significance of the results would be useful for interpreting the results in table 7.

3. The title of the paper is introduced as the main google trend search, but when you look at the results, you can confirm that the technical indicator and keyword are both the main. Therefore, it is recommended to change the paper title.

4. For comparison of results, it would be good to attach the results of a model that did not use both indicators and keywords.

5. What is the meaning of ‘Unknown’ in Table 7.

6. In addition to the results of the roc curve, more accurate results can be obtained by looking at the results of the confusion matrix.

7. Since the returns in figure 6 are very high, additional information is needed to explain this. (Thai stock index graph, etc.)

Round 2

Reviewer 1 Report

Figure 7 is difficult to understand. It is better to show the index graph, first. And then put the graph of Cumulative return for each year. Because the cumulative return for ML model in 2020 is almost 3500%. It is not easy to understand. To understand it, authors should show how much the index is going up at that time. SET100 index, SET index in figure 7 is impossible to identify.

Reviewer 2 Report

Thank you so much for taking my suggestions into considerations. The authors did a good job at improving the manuscript. Below are my comments:

1) The writing of the paper needs to be improved. I was very glad to read that the authors would use the MPDI editing services. I feel uncomfortable with recommending "Accept" prior to this step being completed. The recent additions to the paper need to be copy-edited as well (e.g., line 135, incises instead of indices).

2) Lines 2017-2021 still use the word "investor" instead of "public".
